# Exactly Solvable Models of Interacting Chiral Bosons and Fermions on a Lattice

M. Valiente[*]

Departamento de Física, CIOyN, Universidad de Murcia, 30071 Murcia, Spain
* manuel.v.c@um.es

January 28, 2025

## Abstract

We consider one-dimensional theories of chiral fermions and bosons on a lattice, which arise as edge states of two-dimensional topological matter breaking time-reversal invariance. We show that hard core bosons or their spin chain equivalent exhibit properties that are similar to free fermions, solving the many-body problem exactly. For fermions, we study the effect of a static impurity exactly and show the orthogonality catastrophe in the continuum limit via bosonization. The interacting many-fermion problem in the continuum limit is solved exactly using simple momentum conservation arguments.

## 1   Introduction

Two-dimensional materials subjected to a strong magnetic field exhibit the Quantum Hall effect at low temperatures [1], although the Quantum Hall effect can also occur by breaking time-reversal invariance without magnetic fields, as in Haldane's Chern insulator [2]. This

is a topological effect with explicitly broken time-reversal invariance. As a consequence, one-dimensional chiral modes traversing the edges of the sample emerge and, near the Fermi energy, their energy-momentum dispersion is essentially linear [3]. If one wishes to study edge modes independently of the bulk properties of the material, an effective one-dimensional model consisting of a massless, relativistic fermion can be constructed [3]. Introducing interactions, be it with impurities or among the edge modes themselves, can be done in the momentum representation. However, to study interactions non-perturbatively, the lattice action or, similarly, the lattice Hamiltonian of the theory, is necessarily non-local, due to the Nielsen-Ninomiya theorem [4, 5], which forbids local chiral theories to be placed on a lattice. There are several ways to study chiral particles on a lattice, especially if we abandon the requirement of locality. One such method is called perfect fermions [6], a variation of which has been recently utilized successfully in Quantum Monte Carlo simulations of Helical Luttinger liquids [7]. Another method uses SLAC fermions, whose (periodic) energy-momentum dispersion is perfectly linear, albeit its action is highly non-local [8]. Two branches of SLAC fermions with opposite velocities, as is the case with Helical edge states, have also been investigated recently [9].

All of the above systems and effective lattice models refer to fermions. Naturally, chiral bosons and, in particular, non-interacting chiral bosons do not arise as edge modes in usual topologically non-trivial matter, as its relevant constituents – the electrons – are fermions. However, topological, chiral magnonic (spin-wave) edge modes can exist in a magnet [10, 11]. Therefore, one-dimensional, chiral hard core bosons constitute a potentially realistic system to study within an effective chiral theory.

In this work, we consider one-dimensional, short-range interacting chiral bosons and fermions on a lattice. We accomplish a purely linear energy-momentum dispersion, periodic in the Brillouin zone, by using the SLAC Hamiltonian for the kinetic energy of the particles. We show that, due to the linearity of the dispersion and its periodicity in the Brillouin zone, for multiple particles and at a given total momentum, the non-interacting ground states couple to completely flat bands. In the two-body problem, flat-band scattering theory [12] is used to extract the bound and scattering states exactly. The bosonic many-body problem with zero-range interactions is solved exactly in the hard core limit, which is relevant for chiral spin chains. Its ground state energy is calculated in closed form and coincides with the ground state energy of free fermions, although their wave functions are not related in any simple manner [13, 14]. The pair correlation function and static structure factor are also calculated in closed form, and the momentum distribution is obtained numerically, very easily, thanks to the extremely simple form of the ground state wave function. We also solve fermions interacting with a fixed impurity, both in the fermionic representation on a lattice, as well as in the continuum limit via bosonization [15, 16], and show that both results compare favorably. In the continuum limit, bosonization allows for a one-line proof of the orthogonality catastrophe in this problem [17]. For interacting lattice fermions, the problem does not seem exactly solvable. However, we show that in the continuum limit the non-interacting ground state becomes an eigenstate of the Hamiltonian. We do this by showing that perturbative corrections vanish in the continuum limit. These results coincide with what is obtianed more naïvely using a momentum-space, cutoff regularized theory directly in the continuum limit, as well as with standard bosonization.

## 2   Non-interacting particles

We are going to consider chiral particles on a lattice, whose dynamics, before the inclusion of interactions, are determined by the SLAC Hamiltonian. On a lattice with an odd

number $L_s$ of lattice sites, and under periodic boundary conditions, the single-particle Hamiltonian corresponding with speed $v$ is given by [9]

$$H_0 = 2\hbar v \sum_{\ell=1}^{(L_s-1)/2} J(\ell, L_s) \sin(a\ell\hat{k}), \tag{1}$$

where $a$ ($> 0$) is the lattice spacing, $\hat{k} = -ia^{-1}\partial_\ell$ is the quasi-momentum operator, and

$$J(\ell, L_s) = \frac{(-1)^\ell \pi}{L_s \sin(\pi\ell/L_s)} \tag{2}$$

are the non-local tunneling rates. In the first quantization, the single-particle stationary Schrödinger equation, for each site $\ell$, $H_0 \psi_\ell^{(k)} = E_k \psi_\ell^{(k)}$, is solved by plane waves, $\psi_\ell^{(k)} = \exp(ik\ell a)/\sqrt{L_s}$, with $E_k = \hbar v k$, and the quantization $ka = 2\pi n/L_s$, with $n = -(L_s - 1)/2, \ldots, (L_s - 1)/2$.

The non-interacting ground state for spinless bosons is given by the state in which all bosons occupy the minimum energy state, that is, $\psi^{(k_*)}$, where $k_* a = -\pi(1 - 1/L_s)$. In the second quantization, this state is simply

$$|\psi_0^{(0)}\rangle = \frac{1}{\sqrt{N!}} \left[ b_{k_*}^\dagger \right]^N |0\rangle, \tag{3}$$

where $|0\rangle$ is the vacuum of particles and $b_{k_*}^\dagger$ is the bosonic creation operator with momentum $k_*$. The ground state energy is $E = N\hbar v k_*$. The state in Eq. (3) is not interesting, since every boson occupies a state which, in the continuum limit ($a \to 0$), has infinite momentum and energy, and no meaningful low-energy physics can be extracted from it.

For fermions, the non-interacting ground state is slightly more interesting than for bosons, and corresponds to a filled Fermi sea, that is,

$$|F\rangle = \prod_{k_* \leq k \leq k_F} c_k^\dagger |0\rangle, \tag{4}$$

where the Fermi momentum $k_F = -\pi/a - \pi/(L_s a) + 2\pi\rho$, with $\rho = N/(L_s a)$ the particle density. The ground state energy is obviously

$$E = \sum_{k_* \leq k \leq k_F} E_k = \hbar v \pi \rho (N - L_s). \tag{5}$$

## 3 Interacting bosons

Interactions may change the free-boson picture described above dramatically. In particular, a hard-core on-site two-boson interaction, which makes the model essentially equivalent to a spin chain, also turns the system into an exactly solvable model with a non-trivial ground state and low-energy physics. Hence, we consider an interaction of the form

$$H_\mathrm{I} = \frac{U}{2} \sum_{j=1}^{L_s} n_j(n_j - 1), \tag{6}$$

where $U$ is the interaction strength, and $n_j = b_j^\dagger b_j$ is the number operator at site $j$. The hard core limit corresponds to taking $U \to \infty$.

## 3.1 Two-body problem

Before solving the many-body problem exactly, we consider the two-body problem. In the ground state, the total momentum $K$ is given by $K = 2k_*$. The non-interacting ground state has energy $\hbar v K$, independent of relative momentum. Within the same total momentum sector, there are eigenstates with higher energy. This happens with the SLAC Hamiltonian due to periodicity, unlike the continuum case with hard momentum cutoffs, in which, within the lowest total momentum sector, there is one eigenstate, which is certainly an eigenstate regardless of interactions [12]. The higher-energy, non-interacting eigenstates with the same total momentum (mod $2\pi/a$) have all identical energies, equal to $\hbar v(K + 2\pi/a)$, i.e., they form a flat band. If we define

$$|n_1, n_2\rangle = b^\dagger_{2\pi n_1/L_s a} b^\dagger_{2\pi n_2/L_s a} |0\rangle, \tag{7}$$

then the flat-band states are given by $|n_* + m, n_* + L_s - m\rangle \equiv |m\rangle$ ($m = 1, 2, \ldots, (L_s - 1)/2$), with $n_* = (-L_s + 1)/2$.

Since the two-body interaction preserves total momentum, the reduced one-body problem for $K = 2k_*$ consists of the coupling between $(L_s - 1)/2$ modes in a flat band with energy $\hbar v(K + 2\pi/a)$ and one mode – the non-interacting ground state – with energy $\hbar v K$. The interaction (even considering finite $U$) has zero range. This allows for scattering states at energy $\hbar v(K + 2\pi/a)$ to exist even with finite and odd $L_s$. We use flat-band scattering theory [12], adapted to finite sizes, to obtain all scattering states. If we denote a scattering state (with momentum $K$) with $|\psi_s\rangle$, this satisfies $H_I|\psi_s\rangle = 0$. Writing $|\psi_s\rangle = \sum_{m=1}^{(L_s-1)/2} \psi_s(m)|m\rangle$, we obtain

$$\sum_{m=1}^{(L_s-1)/2} \psi_s(m) = 0. \tag{8}$$

There are $(L_s - 3)/2$ distinct solutions to the above equation. For instance, we can write the (non-orthogonal) scattering states $\psi_s^{(j)}$ as

$$\psi_s^{(j)}(m) = \delta_{m,j} - \delta_{m,j+1}, \; j = 1, 2, \ldots, (L_s - 3)/2. \tag{9}$$

Since all scattering states, regardless of the strength $U$ of the interaction, have zero double occupancy, the interaction couples the ground state with only one state in the flat band. The remaining (unnormalized) state is trivial to calculate by inspection, and is given by $\psi(m) = 1$ ($m = 1, \ldots, (L_s - 1)/2)$). Since neither this state nor the non-interacting ground state are eigenstates of the interacting Hamiltonian, we diagonalize the Hamiltonian in the subspace spanned by these two states. After appropriate normalization of the states, the Hamiltonian in this subspace takes the matrix form

$$H = \begin{pmatrix} \hbar v K + \frac{U}{L_s} & U\frac{\sqrt{L_s-1}}{L_s} \\ U\frac{\sqrt{L_s-1}}{L_s} & \hbar v\left(K + \frac{2\pi}{a}\right) + U\frac{L_s-1}{L_s} \end{pmatrix}. \tag{10}$$

Near the hard core limit, relevant for spin chains and our own discussion, the eigenvalues $E_0$ and $E_+$ are expanded as

$$E_0 = \hbar v\left(K + \frac{2\pi}{L_s a}\right) - \frac{4\pi^2(L_s - 1)(\hbar v)^2}{L_s^2 U} + O(U^{-2}), \tag{11}$$

$$E_+ = U + \hbar v\left[K + \frac{2(L_s - 1)\pi}{L_s a}\right] + O(U^{-1}). \tag{12}$$

Notice that the ground state energy, Eq. (11), is identical to the ground state for two non-interacting fermions when $U \to \infty$. However, their eigenfunctions are not related in a simple manner. The highest excited state of the two bosons in the minimal total momentum sector has an energy given in Eq. (12), which, to leading order, is simply the flat-band energy plus the interaction energy $U$. The ground state for $U \to \infty$ is given by

$$|\psi_0\rangle = \sqrt{\frac{L_s - 1}{L_s}} |\psi_0^{(0)}\rangle - \frac{1}{\sqrt{L_s}} |\psi\rangle, \tag{13}$$

which, in first quantization, in the relative coordinate, and in the position representation, has the following very simple wave function

$$\psi_0(\ell a) = \frac{1}{\sqrt{L_s - 1}} \left[ 1 - \delta_{\ell,0} \right]. \tag{14}$$

That is, the ground state is the non-interacting ground state with the doubly-occupied state removed.

## 3.2 Many-body problem

Here, we consider the many-body problem for interacting bosons. We study the hard core limit ($U \to \infty$), which we show is exactly solvable.

As we shall see below, for hard core bosons, the ground state has an extremely simple form. To study more than two hard core bosons, following the same strategy that solved the two-body problem constructively seems possible – at least for the three-body problem, where we can start with the Faddeev "incident" state [18, 19], which already contains the hard core constraint for one pair at a time. However, Eq. (14) suggests an Ansatz for the $N$-boson ground state of the form

$$\psi_0(x_1, x_2, \ldots, x_N) = A \exp(iK_*X) \prod_{i<j=1}^{N} \left[ 1 - \delta_{\ell_i, \ell_j} \right], \tag{15}$$

where $A$ is the normalization constant, $x_i = \ell_i a$ $(i = 1, 2, \ldots, N)$, $\ell_i = 1, 2, \ldots, L_s$, $N \leq L_s$, $K_* = Nk_*$ and $X = N^{-1}\sum_{i=1}^{N} x_i$ the center of mass coordinate. We prove now that Eq. (15) is indeed the ground state for $N$ hard core bosons, and that the ground state energy is the ground state energy for $N$ non-interacting fermions. For simplicity, and without loss of generality, we set the particle number to $N = N_0 = (L_s + 1)/2$, that is, the half-filled lattice.

To prove that Eq. (15) is the ground state, note that the hard-core Hamiltonian is equivalent to the non-interacting Hamiltonian with no tunneling allowed for particle $j$ from its position to a position occupied by another particle $j' \neq j$. To calculate the ground state energy, we may choose the configuration $\mathbf{x} = (1, 2, \ldots, N_0)a$ and, after inserting the ground state into the stationary Schrödinger equation $(H\psi_0)(\mathbf{x}) = E\psi_0(\mathbf{x})$, we obtain

$$E = \sum_{j=1}^{N_0} \epsilon_j, \tag{16}$$

where $\epsilon_j$ is given by

$$\epsilon_j = -i\hbar v \left[ \sum_{s_j>0}' t(s_j)e^{ik_*s_j} - \sum_{s_j<0}' t(-s_j)e^{ik_*s_j} \right], \tag{17}$$

where the primed sums indicate that they run over all values of $s_j$ such that $x_j + s_j$ is unoccupied. The calculation of the ground state energy reduces, in fact, to a counting problem. Each term containing $\exp(ik_*s)$ appears in Eq. (16) exactly $s$ times (with $s = 1, 2, \ldots, (L_s - 1)/2$). The energy reduces to

$$
E = -\hbar v \sum_{s=1}^{\frac{L_s-1}{2}} \frac{2\pi s}{L_s a} = -\hbar v \pi \frac{L_s^2 - 1}{4 L_s a}, \tag{18}
$$

which is the ground state energy for $N_0$ non-interacting fermions with the SLAC Hamiltonian, as we wanted to show. Choosing any other configuration $(x_1, x_2, \ldots, x_N)$, with $x_i \neq x_j$ for $i \neq j$, yields the same result.

Since the many-boson ground state has a very simple form, some correlation functions can be calculated exactly. The normalization constant $A$ (see Eq. (15)), is given by $A = \prod_{n=0}^{N_0-1}(L_s - n)^{-1/2}$. The pair correlation function, defined via

$$
g(x, x') \propto \sum_{\ell_3, \ldots, \ell_N} |\psi_0(x, x', x_3, \ldots, x_N)|^2, \tag{19}
$$

is given by

$$
g(x, x') = \frac{1}{4}\left[1 + \frac{1}{L_s}\right](1 - \delta_{\ell,\ell'}), \tag{20}
$$

where $x = \ell a$ and $x' = \ell' a$. The static structure factor is calculated immediately as

$$
\begin{aligned}
S(k) &= 1 + \frac{1}{N_0} \sum_{\ell_1, \ell_2} e^{-ik(\ell_1-\ell_2)a}\left[g(x_1, x_2) - \rho^2\right] \\
&= 1 + \frac{1}{4N_0}\left[1 - \frac{1}{L_s}\right] + \delta_{k,0}\left[\frac{1}{4N_0} - \frac{L_s}{4N_0}\right],
\end{aligned} \tag{21}
$$

which, in the infinite-size limit, $L_s \to \infty$, becomes $S(k) = 1 - \delta_{k,0}/2$. As for the momentum distribution, it can be calculated numerically via

$$
\rho(k) = N_0 \sum_{x_2, \ldots, x_{N_0}} \left|\sum_{x_1} e^{-ikx_1}\psi_0(\mathbf{x})\right|^2. \tag{22}
$$

In Eq. (22), we perform the sum over $x_1$ deterministically, while the sum over $x_2, \ldots, x_{N_0}$ is done with a direct Monte Carlo procedure. In Fig. 1 , we observe that the momentum distribution is strongly peaked around $k = k_*$ ($k_* = -\pi/a$ as $L_s \to \infty$), but is clearly non-trivial.

# 4 Interacting Fermions

We now move on to the more standard case of spinless fermions. Obviously, only interactions with non-zero range are non-trivial. The simplest model features nearest-neighbour interactions only, with strength $V_0 > 0$. Before venturing into the interacting many-body problem, we solve and characterize the problem of a static impurity immersed in the Fermi sea.

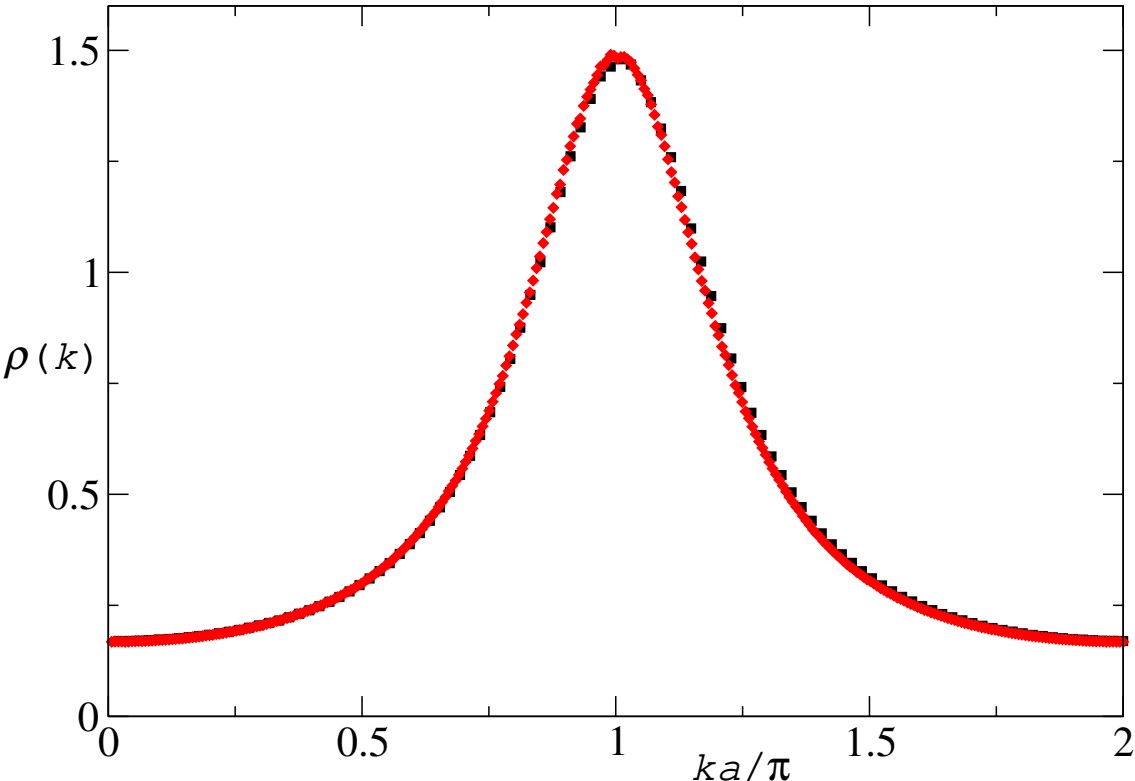

Figure 1: Momentum distribution, calculated using $10^6$ Monte Carlo steps, for hard core bosons at half-filling. Black squares: 51 bosons in 101 sites; red diamonds: 151 bosons in 301 sites. Error bars are smaller than symbol sizes.

### 4.1 Static impurity

The impurity is placed in the center of an otherwise arbitrarily long lattice. Assuming that the impurity-fermion interaction has zero range, in the quasi-momentum representation and in the second quantization, the Hamiltonian is given by

$$H = \hbar v \sum_k k c_k^\dagger c_k + \frac{Ua}{L} \sum_{k,q} c_{k+q}^\dagger c_k, \tag{23}$$

where the sums run over the Brillouin zone $(-\pi/a, \pi/a]$ and momenta are considered $\mathrm{mod}(2\pi/a)$, with $a > 0$ the lattice spacing. Above, $U$ is the interaction strength and $L$ is the length of the system, given by $L = L_s a$, with $L_s$ the number of lattice sites. Using the fermionic representation, the exact solution follows the chiral equivalent of Fumi's theorem [20]. The simplest way to implement it is by simply performing exact diagonalization in the position representation and adding up all single-particle eigenenergies for large enough sizes [19]. Since we are interested in the vacuum of the theory, we shall assume half-filling, that is, $N/L_s = 1/2$ which, in the continuum limit ($a \to 0$), corresponds to a filled Fermi sea with infinite density. The results for the ground state energy are plotted in Fig. 2. Particle-hole excitations correspond to removing a particle from the Fermi sea and placing it above the Fermi point, and the energy of the states is calculated in the exact same manner.

We now compare these results with the results of bosonization. These should agree with each other, if they must, in the continuum limit, that is, for $a \to 0$ or, otherwise, with $a$ fixed, in the lattice weak-coupling regime ($|U|a/\hbar v \ll 1$). The bosonization procedure

is well known [21] and, defining the operators, for $q > 0$

$$\rho_q = \sum_k c_{k+q}^\dagger c_k, \tag{24}$$

with commutation relations $[\rho_q^\dagger, \rho_{q'}] = \delta_{q,q'} qL/2\pi$, we obtain

$$
\begin{aligned}
H \to H_{\mathrm{B}} = {} & \frac{2\pi\hbar v}{L} \sum_{q>0} \rho_q \rho_q^\dagger + \hbar v \sum_{k<k_F} k \\
& + Ua\frac{N}{L} + \frac{Ua}{L} \sum_{q>0} \left[ \rho_q^\dagger + \rho_q \right].
\end{aligned}
\tag{25}
$$

As is clear from the above relation, $H_{\mathrm{B}}$ is not yet diagonalized. However, we may rewrite $H_{\mathrm{B}}$ as follows

$$
\begin{aligned}
\left( \rho_q + \frac{Ua}{2\pi\hbar v} \right) \left( \rho_q^\dagger + \frac{Ua}{2\pi\hbar v} \right) = {} & \rho_q \rho_q^\dagger + \frac{Ua}{2\pi\hbar v} \left( \rho_q^\dagger + \rho_q \right) \\
& + \left( \frac{Ua}{2\pi\hbar v} \right)^2,
\end{aligned}
\tag{26}
$$

and, after defining $\rho_q + Ua/2\pi\hbar v \equiv \tilde{\rho}_q$, we have

$$H_{\mathrm{B}} = \frac{2\pi\hbar v}{L} \sum_{q>0} \tilde{\rho}_q \tilde{\rho}_q^\dagger + E_0, \tag{27}$$

where the ground state energy, as we shall see shortly, is given by

$$E_0 = \hbar v \sum_{k<k_F} k + Ua\frac{N}{L} - \frac{2\pi\hbar v}{L} \sum_{q>0} \left( \frac{Ua}{2\pi\hbar v} \right)^2. \tag{28}$$

Note that the commutation relations for the new operators are preserved, that is, $[\tilde{\rho}_q^\dagger, \tilde{\rho}_{q'}] = \delta_{q,q'} qL/2\pi$. Therefore, we can define bosonic operators $A_q = (2\pi/qL)^{1/2} \tilde{\rho}_q^\dagger$, such that

$$H_{\mathrm{B}} = \hbar v \sum_{q>0} q A_q^\dagger A_q + E_0. \tag{29}$$

To compare, near the continuum limit, the original and bosonized versions of the problem, we begin with the ground state energy. In the bosonized version, setting up an infrared (ultraviolet) cutoff $\Lambda$ (-$\Lambda'$), all we can state is that $E_0 = E_0^{(0)} + g\rho - g^2 \Lambda'/\hbar v(2\pi)^2$, where we have defined the continuum coupling strength $g = Ua$, the density $\rho = N/L$ and the non-interacting ground state energy $E_0^{(0)} = \hbar v \sum_{k<k_F} k$. Therefore, the energy shift contains the Hartree shift $g\rho$ and a divergent term that is negative and quadratic in the coupling $g$. For an effective comparison, we should be able to obtain this term near the continuum limit with lattice fermions. This is easily accomplished using second order perturbation theory on the lattice, which gives

$$E_0 = E_0^{(0)} + g\rho - \frac{\log 2}{2\pi a} g^2 + O((\rho - 1/2a)L^{-1} \log L_s). \tag{30}$$

Clearly, the comparison is favorable. In Fig. 2, we show the ground state energy shift, calculated using exact diagonalization, for a half-filled Fermi gas as a function of the impurity interaction strength, which compares well with the second-order result of Eq. (30) in the weak-coupling regime and up to values of $Ua/\hbar v \approx 1$.

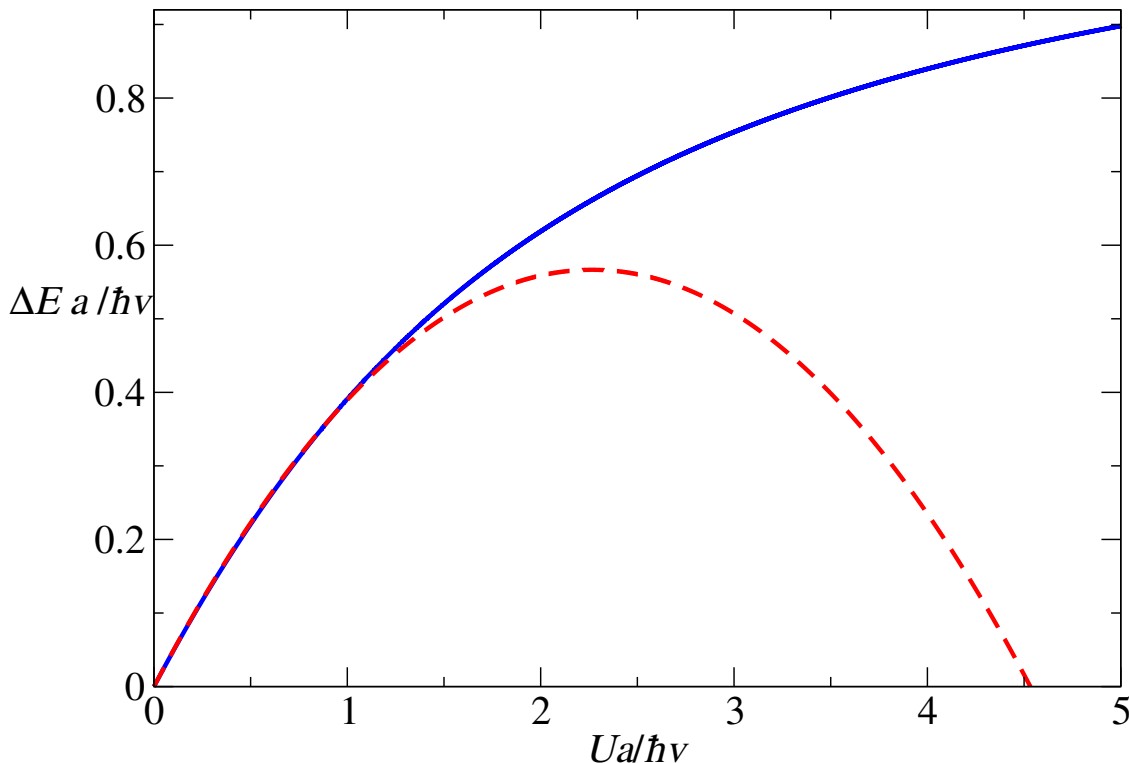

Figure 2: Solid blue line: Energy shift $\Delta E = E_0 - E_0^{(0)}$ in the ground state of a Fermi gas interacting with a fixed impurity as a function of the interaction strength with $L_s = 101$ sites; dashed red line: second order perturbation theory, Eq. (30).

Another interesting feature of the bosonized problem is the ease with which the Orthogonality Catastrophe can be proven: let $|V\rangle$ $(|V_0\rangle)$ be the vacuum with (without) impurity. Then, defining $C(q) = (2\pi/qL)^{1/2}(Ua/2\pi\hbar v)$, we have $\langle V_0|A_q a_q^\dagger|V\rangle = \langle V_0|V\rangle$, which implies that

$$0 = \langle V_0|A_q^\dagger A_q|V\rangle = C(q)\langle V_0|V\rangle, \tag{31}$$

that is, $|V_0\rangle$ is orthogonal to $|V\rangle$, as we wanted to show.

## 4.2 Two-body problem

The two-fermion case is similar to its bosonic counterpart. In the ground state sector, with total momentum $K_* a = -2\pi(1 - 2/L_s)$, there are $(L_s - 3)/2$ flat-band scattering states, which are simple projections onto the zero-interaction manifold [12], and two non-trivial states, namely the ground state and the highest-energy state. The solution follows that in the bosonic problem and is left for the interested reader.

## 4.3 Many-body problem

For finite $V_0$, and before the continuum limit ($L_s \to \infty$ with $a \to 0$, and $L = L_s a$ fixed), the many-fermion problem is not solvable. However, we can show that in the continuuum limit the ground state corresponds to the non-interacting fermionic ground state for any finite $V_0$.

To prove the statement above, it suffices to apply second-order perturbation theory. In fact, a simple yet rigorous estimation that reduces to a counting problem is all we need.

Since the two-body interaction is quartic in fermionic operators and is momentum conserving, second order perturbation theory only couples the ground state to two-particle–two-hole excitations over the Fermi sea. Labelling the non-interacting states by the occupied momentum modes, the ground state $|F\rangle$ has the following quantum numbers $\mathbf{n}_0$

$$\mathbf{n}_0 = (n_*, n_* + 1, \ldots, 0). \tag{32}$$

The excitations that are relevant in this problem are of the form

$$\mathbf{n}(m) = (n_*, n_* + 1, \ldots, n_j + m, \ldots, n_\ell + L_s - m, \ldots). \tag{33}$$

For example, for $m = 1$, there is only one excitation, namely

$$\mathbf{n}(1) = (n_* + 1, n_* + 2, \ldots, -1, 1, (L_s - 1)/2). \tag{34}$$

In general, for each value of $m = 1, 2, \ldots, (L_s - 1)/2$ there are $m^2$ excitations. Therefore, the total number of excitations $N_{\text{ex}}$ that contribute to second order in perturbation theory from the ground state is given by

$$N_{\text{ex}} = \sum_{m=1}^{(L_s-1)/2} m^2 = \frac{1}{6} N(N-1)(2N-1), \tag{35}$$

where $N = (L_s + 1)/2$ is the number of fermions in the Fermi sea. Since the interaction is pairwise, the following bound for the matrix elements holds:

$$|\langle \mathbf{n}(m)|V|\mathbf{n}_0\rangle| \leq \frac{2V_0 a}{L}, \tag{36}$$

and since the coupling is with a flat band, with $E_{\mathbf{n}(m)} - E_{\mathbf{n}_0} = -2\pi\hbar v/a$, we have

$$|E^{(2)}| \leq \frac{V_0^2 a^3}{3\pi\hbar v L^2} N(N-1)(2N-1). \tag{37}$$

In the thermodynamic limit, and at half-filling ($\rho = N/L = 1/2a$), the above second order correction, per particle, becomes negligible, since

$$\frac{|E^{(2)}|}{N} \leq \frac{V_0^2 a}{12\pi\hbar v}, \tag{38}$$

which vanishes in the continuum limit ($a \to 0$), as we wanted to show.

This simple result allows us to confirm that, in the continuum limit, the non-interacting ground state $|F\rangle$ is an eigenstate in the interacting problem. This fact is in agreement with the continuum theory with cutoff regularization [12], in which the only state with minimal momentum $K_*$ is indeed $|F\rangle$ and, therefore, it is an eigenstate of the interaction and the full Hamiltonian. On the lattice, the Hartree shift in the thermodynamic limit (but with $a$ fixed) is calculated as

$$\langle F|V|F\rangle = \frac{L}{8\pi^2} \int_{-\pi/a}^{0} dk \int_{-\pi/a}^{0} dk' \left[ V(0) - V(k' - k) \right], \tag{39}$$

and, using $V(k - k') = 2V_0 a \cos[(k - k')a]$ for nearest-neighbor interactions, we obtain

$$\frac{\langle F|V|F\rangle}{N} = \frac{LV_0}{N4\pi^2 a} \left[ \pi^2 - 4 \right] = \frac{\pi^2 - 4}{2\pi^2} V_0, \tag{40}$$

where $\rho = N/L$ is the density of the filled Fermi sea, which diverges in the field-theoretical or continuum limit $a \to 0$.

We compare now these results to those of the continuum theory with cutoff regularization. In that case, the simplest effective two-body interaction – in one-to-one correspondence with nearest-neighbor interactions on a lattice – is given by [19]

$$V(k', k) = -g_1 q^2, \tag{41}$$

corresponding with the leading order (LO) interaction for fermions in pionless effective field theory (EFT). To compare with the lattice theory, we should rewrite the bare coupling $g_1 = \lambda V_0 / \Lambda^3$, where $\lambda$ is a dimensionless constant, and $\Lambda$ is the infrared momentum cutoff of the theory. The energy shift takes the form

$$\frac{\langle F|V|F \rangle}{N} = \frac{\lambda V_0}{48\pi^2} \frac{\Lambda}{\rho} = \frac{\lambda}{24\pi} V_0, \tag{42}$$

where we have used the infrared cutoff-density relation, $\Lambda = 2\pi\rho$. This results in the value for the continuum coupling constant $\lambda = 12(\pi^2 - 4)/\pi$.

We can also compare the continuum exact results above with the results obtained via bosonization. We shall be careful to not drop any constant terms that may be compared with the exact results using fermions. We begin with the interaction $V$, with the LO-EFT parametrization, Eq. (41). In the continuum limit, we have

$$V = \frac{1}{2L} \sum_{kk'q} V(q) c^\dagger_{k+q} c^\dagger_{k'-q} c_{k'} c_k, \tag{43}$$

which we prepare for bosonization, by rewriting it as

$$\begin{aligned}
V &= -\frac{1}{2L} \sum_{kk'} V(k'-k) c^\dagger_{k'} c_{k'} + \frac{1}{2L} \sum_{kk'q} V(q) c^\dagger_{k+q} c_k c^\dagger_{k'-q} c_{k'} \\
&= \frac{g_1}{4\pi} \sum_k \left( \frac{2\Lambda^3}{3} + 2k^2\Lambda \right) c^\dagger_k c_k + \frac{1}{2L} \sum_{q>0} V(q) \sum_k c^\dagger_{k+q} c_k \sum_{k'} c^\dagger_{k'-q} c_{k'} \\
&\quad + \frac{1}{2L} \sum_{q>0} V(q) \sum_k c^\dagger_{k-q} c_k \sum_{k'} c^\dagger_{k'+q} c_{k'} + \frac{V(0)}{2L} \left[ \sum_k c^\dagger_k c_k \right]^2.
\end{aligned} \tag{44}$$

The last term above vanishes, since $V(0) = 0$ in LO-EFT. The first term in Eq. (44) can be substituted with a c-number, since we are sure that the ground state is just the filled Fermi sea. This is given by

$$\frac{g_1}{4\pi} \sum_k \left( \frac{2\Lambda^3}{3} + 2k^2\Lambda \right) \langle c^\dagger_k c_k \rangle = \frac{g_1 L}{8\pi^2} \int_{-\Lambda}^0 dk \left( \frac{2\Lambda^3}{3} + 2k^2\Lambda \right) = \frac{g_1 L \Lambda^4}{6\pi^2} = \frac{\lambda L \Lambda V_0}{6\pi^2}. \tag{45}$$

The rest of the interaction is directly bosonizable, and we obtain

$$\begin{aligned}
V &= \frac{\lambda L \Lambda V_0}{6\pi^2} + \frac{1}{2L} \sum_{q>0} V(q) \left[ \rho_q \rho^\dagger_q + \rho^\dagger_q \rho_q \right] \\
&= \frac{\lambda L \Lambda V_0}{6\pi^2} + \frac{1}{2\pi} \sum_{q>0} q V(q) \left[ a^\dagger_q a_q + \frac{1}{2} \right].
\end{aligned} \tag{46}$$

The kinetic energy is bosonized as usual [15], and becomes

$$H_0 = \hbar v \sum_q q a^\dagger_q a_q + \mathcal{C}, \tag{47}$$

where $\mathcal{C}$ is an infinite constant given by the energy of the non-interacting Fermi sea, that is

$$\mathcal{C} = \hbar v \frac{L}{2\pi} \int_{-\Lambda}^{0} dk\, k = \frac{\hbar v \Lambda^2 L}{4\pi}. \tag{48}$$

Putting it all together, we get, for the (infinite) ground state energy $E_B$ from bosonization, with respect to the non-interacting Fermi gas,

$$E_B = \frac{\lambda L \Lambda V_0}{6\pi^2} - \frac{g_1 L}{8\pi^2} \int_0^{\Lambda'} dq\, q^3 = \frac{\lambda L \Lambda V_0}{6\pi^2} - \frac{g_1 L \Lambda'^4}{32\pi^2} = \frac{\lambda L \Lambda V_0}{6\pi^2} - \frac{\lambda V_0 L \Lambda'}{32\pi^2}. \tag{49}$$

Above, $\Lambda'$ is the ultraviolet cutoff, which must satisfy $\Lambda'/\Lambda \to c$, with $c$ a finite constant, as the infrared and ultraviolet cutoffs are removed. Setting $\Lambda' = c\Lambda$, and using $\Lambda = 2\pi\rho$, we obtain

$$\frac{E_B}{N} = \lambda V_0 \frac{16 - 3c}{48\pi}. \tag{50}$$

For the bosonization result to agree with the exact result, we must demand $c = 14/3$. This finishes the ground state comparison.

Regarding the gapless excitations, the bosonized Hamiltonian gives straight away an excitation dispersion of the form

$$\epsilon_B(q) = \hbar v q \left[ 1 + \frac{V(q)}{2\pi\hbar v} \right]. \tag{51}$$

However, since $V(q) = -g_1 q^2$ scales as $\Lambda^{-3}$, the low-energy excitation spectrum remains unchanged, and identical to free fermions, as $\Lambda \to \infty$. This is exactly what happens, as well, in the exact fermionic treatment and we have, as $\Lambda \to \infty$,

$$\epsilon(q) = \epsilon_B(q) = \hbar v q. \tag{52}$$

We have now shown that, if special care is taken when dealing with regularized expressions, one branch of interacting chiral fermions on the lattice, in the continuum limit, agree with straightforward, constructive bosonization.

## 5    Conclusions and outlook

In this article, we have studied chiral bosons and fermions with short-range interactions, placed on a lattice with purely linear dispersion, using the so-called SLAC Hamiltonian. We have solved a number of models exactly, including hard core bosons on a lattice, fermions interacting with an impurity and the continuum limit of the many-fermion problem. For fermions, all the results compare well with the corresponding solution using bosonization in the continuum limit. In this context, we have given a very simple proof of the orthogonality catastrophe. These results are relevant for interacting problems of edge states in topologically non-trivial many-body systems.

Our results may be useful for studying either soft core, zero-range interacting chiral bosons or, more interestingly, as a starting point for more interesting interacting spin-chains beyond the hard core constraint only. Other methods to regularize and renormalize the effective theories, such as perfect lattice actions, could pave the way towards studying these problems, including intrabranch interactions, when the edge states are helical, as was recently done in Ref. [7]. Moreover, effects of the interplay between non-magnetic impurities and the bulk, which are not necessarily non-trivial [22, 23], may be included by coupling the models we have studied in this work with an in-gap resonant state that appears even in the vacancy limit.

## Funding information

This work was supported by the Ministry of Science, Innovation and Universities of Spain through the Ramón y Cajal Program (Grant No. RYC2020-029961-I), and the national research and development grant PID2021-126039NA-I00.

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
