# Peer review of "Exactly Solvable Models of Interacting Chiral Bosons and Fermions on a Lattice"

_SciPost Physics_

## Round 1 · Referee Report · Anonymous (Referee 1) · 2025-3-14

Strengths

see below

Weaknesses

see below

Report

The idea to consider particles on a lattice with strictly linear dispersion analytically in momentum space (both fermions and bosons) is an interesting idea. It maybe useful for understanding the fundamental differences (similarities) between continuous and discrete theories, and give more benchmarks to test against and even help the numerical techniques by giving more insights about the structure of the wave functions.

While I do find this work suitable for SciPost Physics, there are issues that I would hope the authors can address.

  1. It may be obvious to some, but it is not explained why the total momentum K=2k_*. And since the whole derivation of the following relies on it, I think it should be explained properly.

  2. What is actually the form of the interaction that is considered for the interacting fermions? Later in the section they calculate matrix elements of operator V. It’s impossible to check the answers because it’s not very clear what is the form of the operator V that is being substituted there.

  3. Before (31) a_q^\dagger was never defined. The transition in the second equality of (31) is not clear, I do not understand it, it should be explained.

  4. “To prove the statement above, it suffices to apply second-order perturbation theory” - Why is it possible to use perturbation theory in 1d interacting fermionic case? It counters the fact that all the interaction effects for fermions in 1d are non-perturbative.

  5. The form of states (33) and (34) is not really clear. What is l? What is j? A better explanation is needed for these notations.

  6. The (36)-(37) transition I do not understand, should there be at least a reference to something that shows how to do such derivations?

  7. As far as I understand the whole calculation of 32-38 was supposed to show that there is no difference in ground state energy and therefore no difference in the ground state. Is it a clear implication? Moreover later in 39-42 the Hartree shift, which is a shift in energy due to interaction (assuming that the ground state is the same), is calculated again, and here it is not equal to zero in the continuum limit. Does it mean that the whole derivation of 32-38 was wrong? In short, I do not find results (38) and (40),(42) consistent. One says that there is no change in energy in the continuum limit, the other gives the finite value of the shift.

  8. The whole derivation of (42) is not clear. There is not even a single reference to all these seemingly very sophisticated techniques and facts mentioned in the paragraph above (42). Or do they mean that everything is explained in [19]?

  9. What is \epsilon(q) in (52), it was not defined and calculated before, wasn’t it? So what does this comparison of \epsilon(q) and \epsilon_B(q) mean then?

Requested changes

see above

Recommendation

Ask for major revision

---

## Round 1 · Referee Report · Anonymous (Referee 2) · 2025-6-10

Strengths

see report

Weaknesses

see report

Report

This manuscript studies the Bosonic and Fermionic models with linear dispersion and local interaction on a lattice. For the bosonic mode, the authors studied the exact many-body ground state at the strong coupling limit (U \to \infty). They show that in the strong coupling limit, the bosonic many-body state is identical to a non-interacting Fermionic model.

For the Fermionic model, the authors initially examined an impurity model (a linearized fermion with a delta potential) and solved it via bosonization. Then they studied this model in the continuum limit as a\to0. In this limit, the model can be mapped to a Free boson theory via bosonization.

Overall, this manuscript presents an interesting model within specific limits, and I think this work can be published in SciPost Physics after explicitly specifying the novelty of their work and addressing the following questions: 1. For the bosonic model, I don’t believe this model is integrable at any finite interaction strength U, and it would be nice if the authors could add a discussion on this point. 2. For the Fermionic model, the authors solved a linearized fermion in a continuum with contact interaction. This model is known as a Luttinger liquid, which can be mapped to a free boson field theory via bosonization. Can the authors specify the Luttinger parameter for their lattice model as a function of the interaction strength? And a comment: At infinite repulsive coupling, for any dispersion of bosons model, it should be the same as a free Fermionic model, since the only difference between bosons and fermions is that you can populate more than one boson at a site, but not for fermions. Can you generalize the result for a generic dispersion? Since we know this result for quadratic dispersion from Tonks-Girardeau gas, what is the novel aspect of the work, provided that at the infinite coupling limit, it should work for any dispersion?

Recommendation

Ask for minor revision

---

## Editorial Decision

in_refereeing